# Autonomous Visual Navigation System Based on a Single Camera for Floor-Sweeping Robot

**Jinjun Rao ***, **Haoran Bian, Xiaoqiang Xu and Jinbo Chen**

School of Mechanical Engineering and Automation, Shanghai University, Shanghai 200444, China
* Correspondence: jjrao@shu.edu.cn; Tel.: +86-13661938797

**Abstract:** The indoor sweeping robot industry has developed rapidly in recent years. The current sweeping robot environment perception sensor configuration is more diverse and generally does not have active garbage detection capabilities. Advances in computer vision technology, artificial intelligence, and cloud computing technology have provided new possibilities for the development of sweeping robot technology. This paper conceptualizes a new autonomous visual navigation system based on a single-camera sensor for floor-sweeping robots. It investigates critical technologies such as floor litter recognition, environmental perception, and dynamic local path planning based on depth maps. The system is applied to the TurtleBot robot for experiments, and the results show that the mAP accuracy of the autonomous visual navigation system for fine trash recognition is 91.28%; it reduces the average relative error of depth perception by 10.4% compared to conventional methods. Moreover, it has greatly improved the dynamics and immediacy of path planning.

**Keywords:** floor-sweeping robot; visual navigation system; floor litter recognition; depth perception; path planning; superpixel segmentation

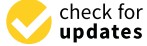



## 1. Introduction

The huge market potential of the floor-sweeping robot has attracted the attention of researchers. The existing sweeping robots have various structures, which are generally divided into outdoor and indoor products. The outdoor type is generally a large commercial robot used to weed and clean the road [1], while the indoor type is a small household robot used to clean garbage and dust [2]. According to most studies, the sweeping robot is able to navigate and plan a path [3,4]. In order to enable all robots in the system to safely bypass obstacles in the environment to reach the target position, obstacle avoidance control is required [5]. Vision technology has made sweeping robots develop active garbage detection and recognition, sense the environment and establish maps, plan the sweeping path, and actively avoid obstacles [6,7]. This actually benefits from the use of industrial cameras, which are small in size, light in weight, low in energy consumption, and have a wide field of view [8–10].

Additionally, in order to reduce costs and improve cleaning efficiency, some researchers use multiple robots for collaborative cleaning, including single-task cleaning [11,12] and multi-task cleaning [13–17]. In the multi-task cleaning work, the tasks need to be scheduled and assigned to each robot in the cluster. In the process of collaborative cleaning, it is necessary to plan the motion of the robot cluster. Their motion planning strategies are divided into offline planning [11] and real-time planning [12]. Some metaheuristic algorithms are also applied in clusters [18,19] to attain global optimization, so real-time planning is more deterministic than offline planning. Most of these sweeping robots carry out global coverage path planning in a simple environment. Even if the distributed task configuration is adopted [16,17], most of the experiments use isomorphic robots. The high computation cost of these robots limits their work efficiency. Therefore, it is necessary to reduce the consumption of computing resources while improving the functional stability of single-robot autonomous perception, modeling, navigation, obstacle avoidance, etc.

In order to enable the sweeping robot to work cooperatively with other full coverage light sweeping robots in the factory environment, this paper proposes a new autonomous vision navigation system of a sweeping robot based on a single-camera sensor. The system uses wireless communication and cloud computing to reduce application costs, and the key technologies in the system's application are studied, such as ground waste identification, environment awareness, and dynamic local path planning based on a depth map. Further, we carried out experiments to verify the system's performance.

The following sections are organized as follows: in Section 2, the framework of autonomous visual navigation system is presented; in Section 3, the key technologies of the autonomous visual navigation intelligent system are studied, including garbage identification and target detection (Section 3.1), environmental depth information perception (Section 3.2), and dynamic local path planning algorithm based on superpixel segmentation (Section 3.3); experiments were conducted in Section 4, including garbage target detection experiment (Section 4.1), indoor depth perception, and dynamic local planning experiment (Section 4.2); the conclusion and outlook are given in Section 5.

## 2. Framework of Autonomous Visual Navigation System

On account of the unique advantage of the camera in visual navigation, we conceptualized a single-camera-based sweeping robot system using cloud computing, as shown in Figure 1. Since the system used wireless transmission and a cloud computing model, it did not require additional high-performance computing resources. As a result, it reduces the hardware cost and provides the possibility of implementing cloud control of the robot.

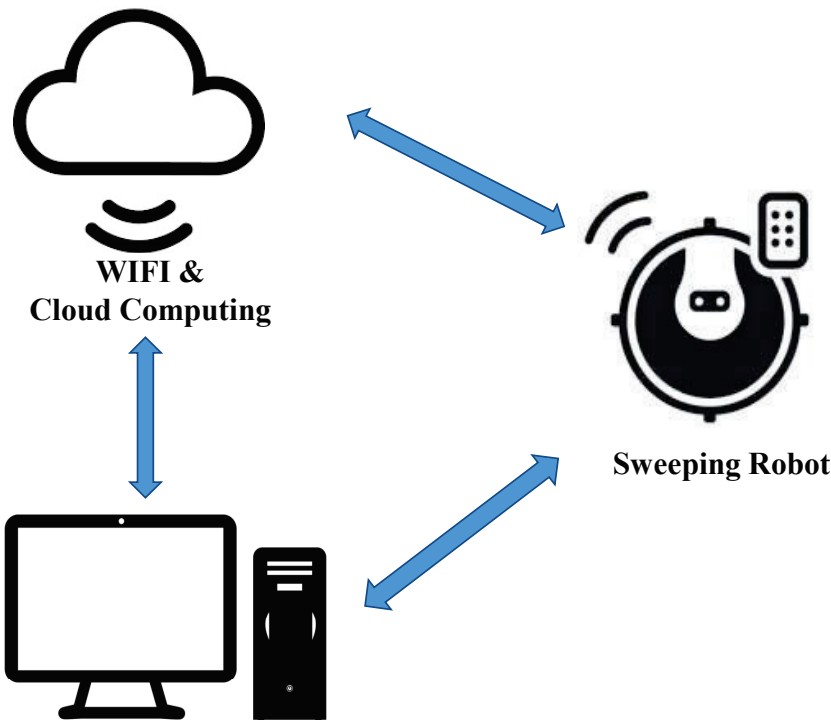

**Figure 1.** Sweeping robot using cloud computing.

The system will significantly reduce the number and type of sensors for the new sweeping robot. The full utilization of single-camera visual information is the key to achieving an autonomous visual navigation system for the robot. This autonomous visual navigation system needs to have at least the following three functions:

(1) Garbage recognition can determine the relative orientation between the robot and the garbage.
(2) Estimation of the distance between the robot and furniture under monocular vision can be used for indoor environment perception of the robot.
(3) Distinguishing the obstacle areas from the clear areas forms the basis for dynamic local path planning.

For the above three system functions, we proposed the technical process shown in Figure 2, which formed the basic technical framework of the autonomous visual navigation system.

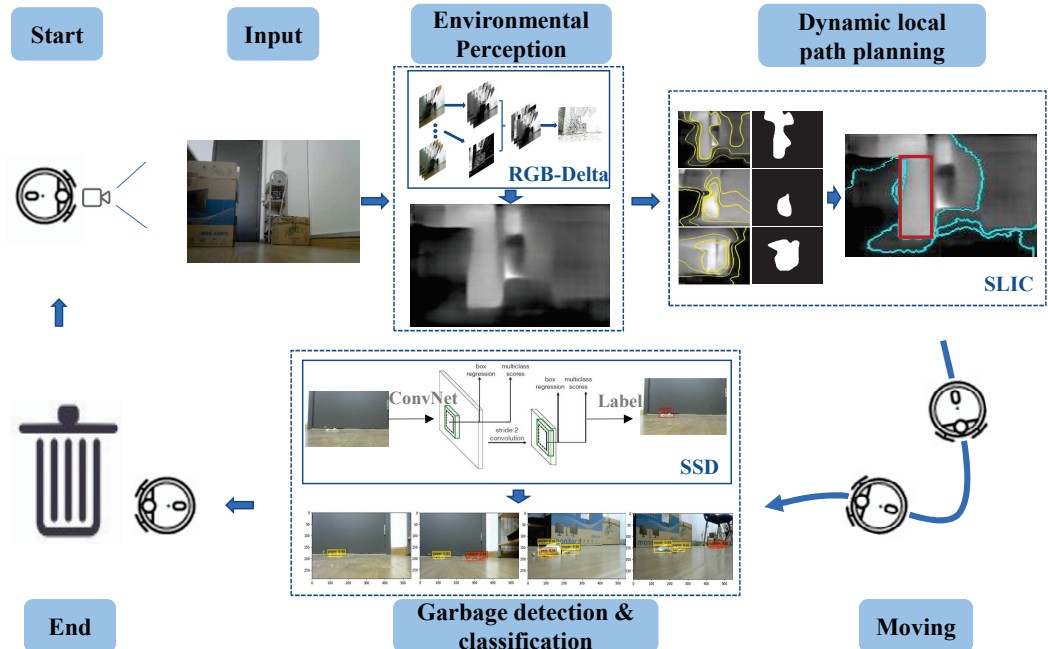

**Figure 2.** The workflow of the autonomous visual navigation sweeping robot.

In this framework, the target detection algorithm SSD [20] (Single Shot MultiBox Detector) is used to identify garbage to actively avoid blind cleaning. To achieve distance perception between the robot and obstacles, we proposed an improved method based on a self-encoder fully convolutional neural network to transform the environment perception problem into a monocular vision depth estimation problem. Based on the environment depth estimation, the depth information map is partitioned into feasible and infeasible regions using the superpixel segmentation algorithm to guide the indoor dynamic local path planning. We only require the image information from a single camera and use intelligent technologies such as deep neural networks in our solution. It utilizes cloud computing and centralized computing to reduce the production cost of the sweeping robot while also extending functions such as cloud monitoring and cloud cleaning. The workflow of the autonomous vision navigation sweeping robot is shown in Figure 2.

In Figure 3, the process of the autonomous visual navigation method of the sweeping robot based on superpixel segmentation can be summarized in the following steps:

(1) Inspection state. The robot rotates the direction randomly to sample images and detect garbage.
(2) Targeting. After detecting the garbage, the robot adjusts its direction so that the cleaning target is in the robot's view.
(3) Environment perception. The robot performs environmental depth perception based on its position and calculates the distance from the robot to the obstacles and the distance from the robot to the targets.

(4) Dynamic local path planning. The robot uses a superpixel image segment algorithm to process the image, determine the through and obstacle areas in the depth information map, and plan the way forward.

(5) Directional calibration. As the robot moves forward, it continuously acquires images, re-plans the next step forward, and calibrates the forward direction in time.

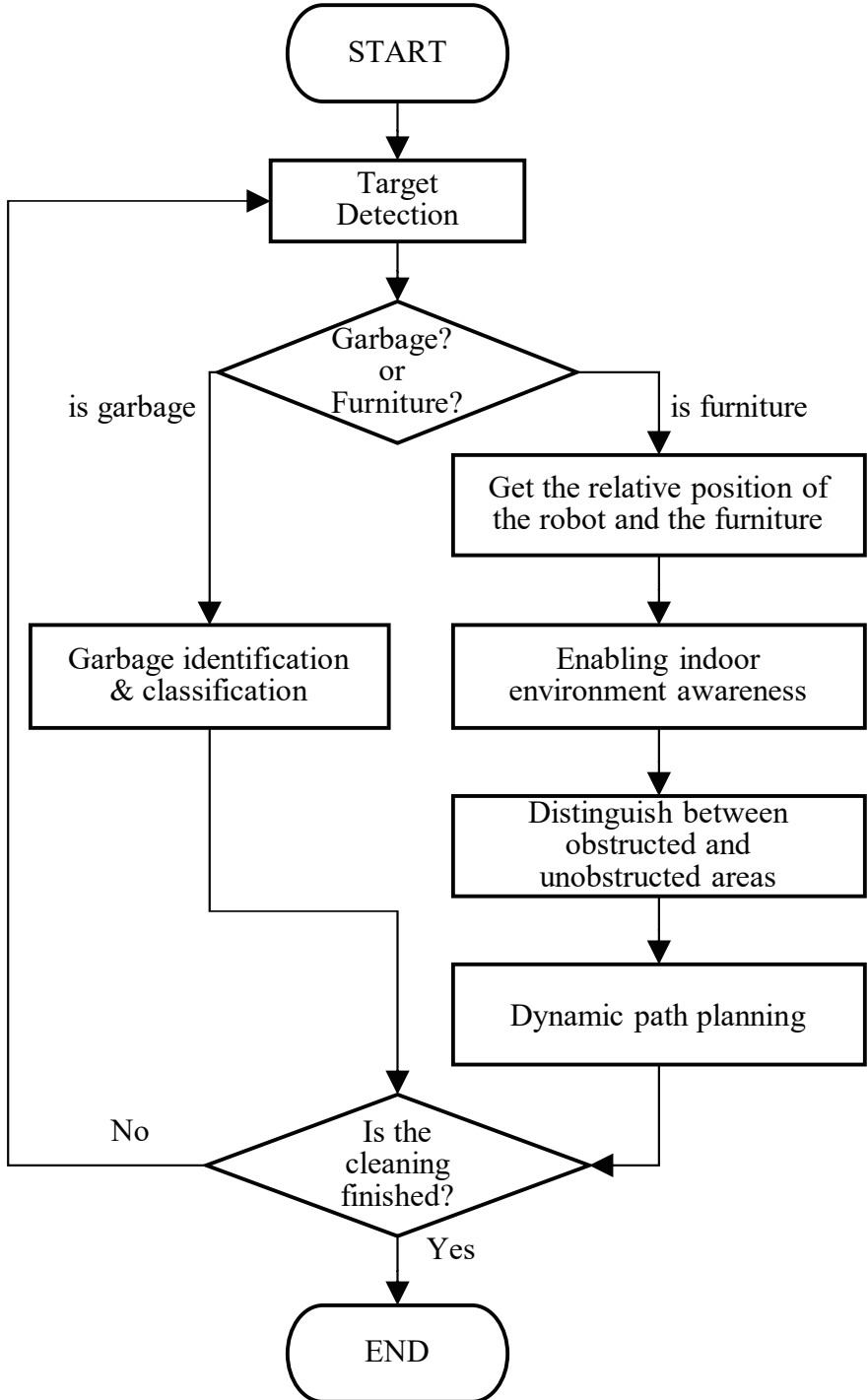

**Figure 3.** Autonomous visual navigation method for sweeping robot based on superpixel segmentation.

## 3. Key Technologies

### 3.1. Garbage Identification and Target Detection

Different from the garbage in the street, ground garbage is mostly paper scraps, dust, and wrapping paper. The small size, inconspicuousness and scattered distribution of indoor

garbage make it more difficult to identify. Traditional target detection methods use manual extraction of image feature information, which cannot avoid incomplete feature information in this process, thus leading to poor recognition results. In contrast, neural networks and deep learning can solve these problems well. Mahankali et al. [21] proposed a new system to identify illegal dumping in mobile vehicles by using video analysis, which is based on the Haar cascade classifier, background subtraction, and OpenALPR library to identify illegal dumping in a changing light environment. Liu et al. [22] used target box dimensional clustering and classification network pre-training to improve the YOLOv2 network model for spam detection and ported it to an embedded module to achieve lightweight features. Sousa et al. [23] proposed a hierarchical deep-learning method for waste detection and classification in food trays. It takes advantage of Faster R-CNN to achieve support for waste classification tasks in high-resolution bounding boxes. Setiawan et al. [24] used the SIFT algorithm to extract labels to recognize the organic and non-organic elements. This paper uses a target detection algorithm based on deep learning techniques. Several familiar image processing methods are shown in Table 1. The deep learning-based target detection algorithm has a significant advantage in accuracy and better generalization performance.

**Table 1.** Comparison of performance characteristics of several image-processing methods to detect a target.

| Processing Method | Precision | Accuracy | Data Requirements | Computational Resource Consumption |
|---|---|---|---|---|
| CCM [1] [25,26] | Low | Low | Low | Low |
| MRF [27,28] | Relatively low | General | Low | Low |
| HOG+SVM [29] | High | High | Relatively high | General |
| Deep Learning [30,31] | Relatively high | Relatively high | Very high | Very high |

[1] CCM is Connected Component Marking.

In our research, the target detection SSD algorithm [20] was used to detect garbage and achieve the active localization of ground targets. SSD used a CNN for detection, which employed a multi-scale feature map. It used VGG16 [32] as the base model, eliminating the fully connected layers and adding multiple convolutional layers. The core of the SSD algorithm includes the use of multi-scale feature maps, convolutional neural networks, and setting prior frames to apply to the detection of small targets.

### 3.2. Environmental Depth Information Perception

The ability of a sweeping robot to adapt to its working environment depends on its appropriate understanding of the environment. In this section, the self-encoder fully convolutional neural network algorithm proposed by Laina et al. [33] is improved to supervise the network learning by introducing a parallax map and using the parallax information between the upper and lower frames of the image sequence input to the neural network.

Parallax maps are highly correlated with depth in 3D information, and many studies of stereo visual depth information have fully exploited parallax information [34]. According to the principle of optical field imaging, the geometric relationship between parallax and depth can be expressed by Equation (1):

$$\Delta x = -\frac{f}{z}\Delta u, \tag{1}$$

where $f$ is the camera's focal length, $z$ is the depth value of the P-point, $\Delta x$ denotes the displacement of the P-point at different viewpoints (parallax), and $\Delta u$ is the baseline length of the camera lens array.

Due to the solid geometric correlation between the parallax information and the depth information, we adopt the integration of parallax information into the fully convolutional neural network. The calculation method of the parallax map has been researched in many ways. We take the simplest way to calculate it, the absolute difference method, as shown in Equation (2).

$$DA = |f(x_l) - f(x_r)|,\qquad(2)$$

where $f(x_l)$, $f(x_r)$, and $DA$ represent the luminance values of the previous frame, the next frame, and the corresponding absolute parallax map, respectively. The absolute difference map is not a parallax map in the strict sense, but its calculation is fast and straightforward. The calculation result has reference significance, which meets the requirement of using parallax information as the auxiliary input information of the fully convolutional neural network. The parallax map and the original scene image are shown in Figure 4.

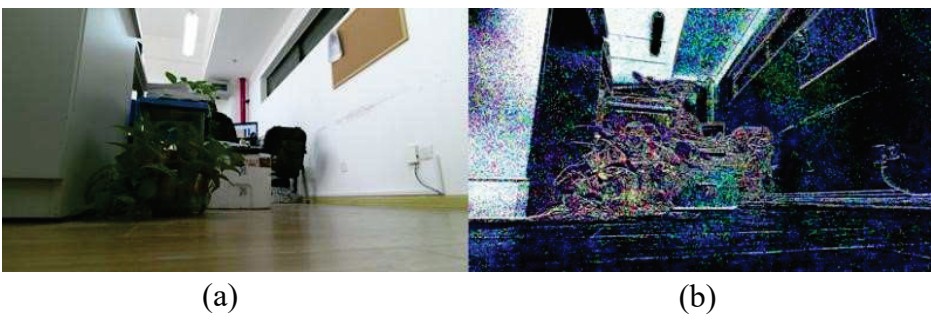

(a) (b)

**Figure 4.** Original image and parallax chart of the indoor scene: (**a**) Original image (**b**) parallax map.

The pseudo-code of the algorithm is shown in Algorithm 1. The grayscale map of parallax information is used as the fourth channel Delta data, fusing with the original RGB three-channel image of the scene to generate RGB-Delta data. These data are then fed into the neural network instead of the original RGB image, as Laina et al. [33] did. The processing flow is shown in Figure 5.

---

**Algorithm 1** RGB-Delta image data fusion.

---

**Require:** *Image from previous frame $img_a$, next frame $img_b$*
**Ensure:** *Four channel $RGB - Delta$ image $img_f$*
1: *Extraction the parallax map between frames $img_{dis} = |img_a - img_b|$*
2: *Initializes a four−dimensional array $img_r$ of the same size as $img_a$*
3: *The 1st dimension of $img_f$ = the R channel of $img_a$*
4: *The 2nd dimension of $img_f$ = the G channel of $img_a$*
5: *The 3rd dimension of $img_f$ = the B channel of $img_a$*
6: *The 4th dimension of $img_f$ = $img_{dis}$*
7: **return** $img_f$

---

### 3.3. Dynamic Local Path Planning Algorithm Based on Superpixel Segmentation

To guide the robot to remove garbage precisely, we proposed a dynamic indoor path planning method for sweeping robots based on superpixel segmentation. The method dynamically processes the depth map in the image sequence, continuously extracts the information of the non-obstacle area, and plans the forward direction. This method is simple in calculation, fast in processing, and can correct the forward movement in time, avoiding producing long-term accumulated errors and improving path planning efficiency.

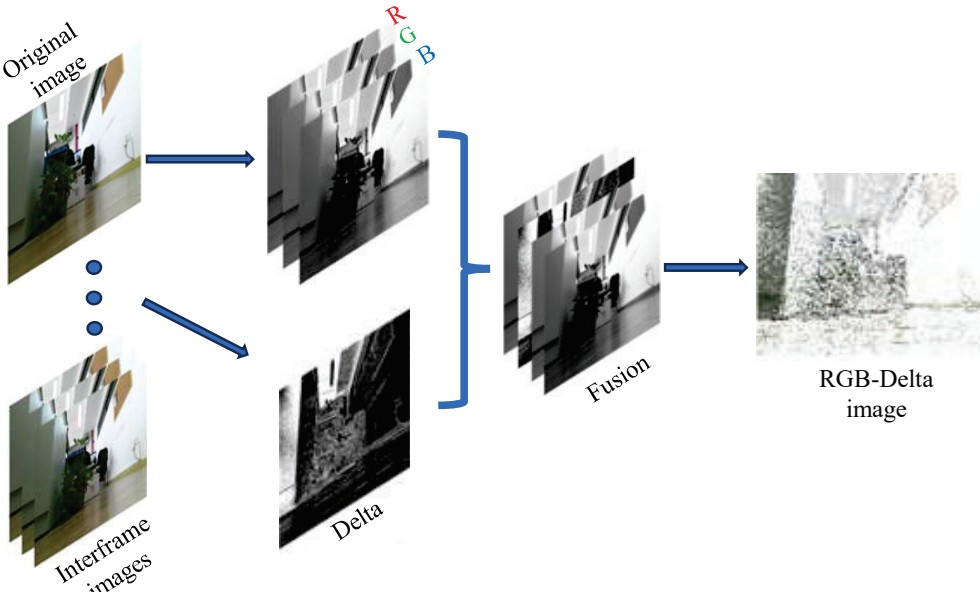

**Figure 5.** RGB-Delta image fusion process.

The environmental depth information is shown in Figure 6. The depth map contains the distance information of the scene in Figure 6b, which shows:

(1) Obstacle areas have similar depth information performance due to their spatial relationship so that they can be fitted into superpixel blocks well after superpixel image segmentation.

(2) Non-obstacle areas have the largest depth information values and are highly similar in terms of numerical relationships. Therefore, it is easy to cluster into the same superpixel block in superpixel segmentation processing.

(3) Objects in different depth environments are divided into different superpixel blocks, and adjacent superpixel blocks are delimited clearly. The relative depth relationships of various objects match well when comparing the images captured in real environments.

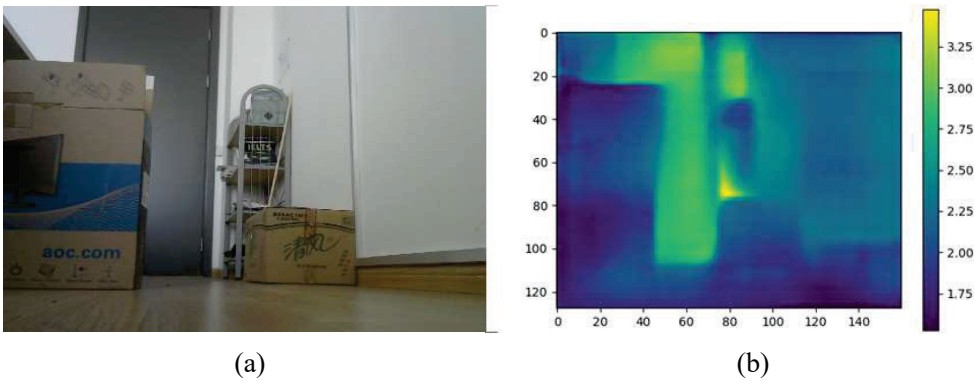

(a)                                                         (b)

**Figure 6.** Indoor environment with its depth information; (**a**) Original scene image, (**b**) Depth map.

We used the SLIC (Simple Linear Iterative Clustering) [35] superpixel image segmentation algorithm to process the depth images. Based on the segmentation results, which are obtained by the superpixel dynamic local planning algorithm, the non-obstacle areas can be judged as optional forward directions.

After the superpixel segmentation algorithm processes the depth map, the pixel values within each superpixel block are similar in size to represent the same area or object. The

distribution of the non-obstacle areas in the image can be obtained from the average pixel value within the superpixel block, which can be calculated according to Equation (3):

$$Value_{unobstacle} = max(\frac{1}{N}\sum_{i=0}^{N} S_j),$$

(3)

where $S_j$ denotes the jth superpixel block and *i* denotes the pixel points within it. For the depth map after the superpixel segmentation process, the pixel average within each superpixel block is calculated. The one with the largest average value means the region with the largest distance in space, which is the unobstructed region, and the calculation result is shown in Figure 7d.

Figure 7a indicates the furthest light gray room door and the direction leading to the non-obstacle area. Figure 7b shows the depth information estimated by the indoor depth information sensing module. Figure 7c shows the segmentation results using the SLIC superpixel algorithm, and the farthest area from the camera is accurately segmented. Based on the principle of the maximum pixel mean value of the superpixel block, we segmented and judged the depth map and extracted the area with the maximum mean value, as shown in Figure 7d. It shows that the judgment of the non-obstacle area by the algorithm matches the actual 3D scene.

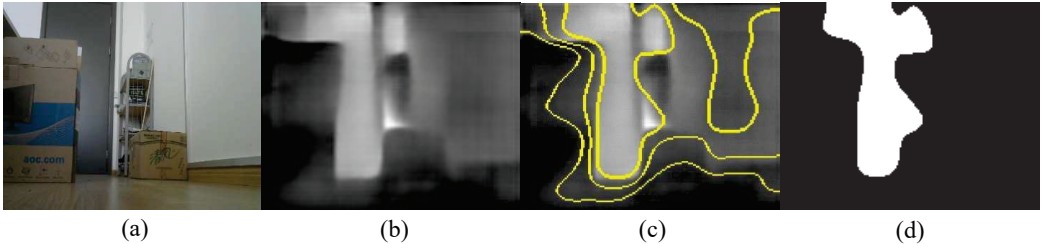

(a)             (b)             (c)             (d)

**Figure 7.** Superpixel segmentation and non-obstacle area extraction based on the depth map: (**a**) Original scene image, (**b**) Depth map, (**c**) Segmentation result, (**d**) Non-obstacle area.

## 4. Experiments

The sweeping robot experiment uses the Turtlebot2 experimental platform, as shown in Figure 8, which is a tiny, low-cost, ROS-based mobile robot. The camera of this experimental platform captures images and transmits the image information to the cloud server through the Wi-Fi module. The signal range of the Wi-Fi is within 40 square meters, and the signal delay is within 50 ms. The result of the cloud calculation is returned to the microprocessor through the Wi-Fi module. The microprocessor generates instructions based on the calculation results, and the robot starts to perform the task. The hardware and software used for the experiments are shown in Tables 2 and 3.

**Table 2.** Experimental hardware environment.

| Hardware | Description |
| --- | --- |
| Wi-Fi | TP-LINK |
| Micro-processor | SZ02-232-2K, S202-NET-2K |
| Camera | Logitech C270c |
| GPU | NVIDIA GTX 1080TI Single video memory 11G, memory 32G |
| CPU | AMD Ryzen 5 5600X 6-Core Processor 3.70 GHz |
| Platform | Turtlebot2 |

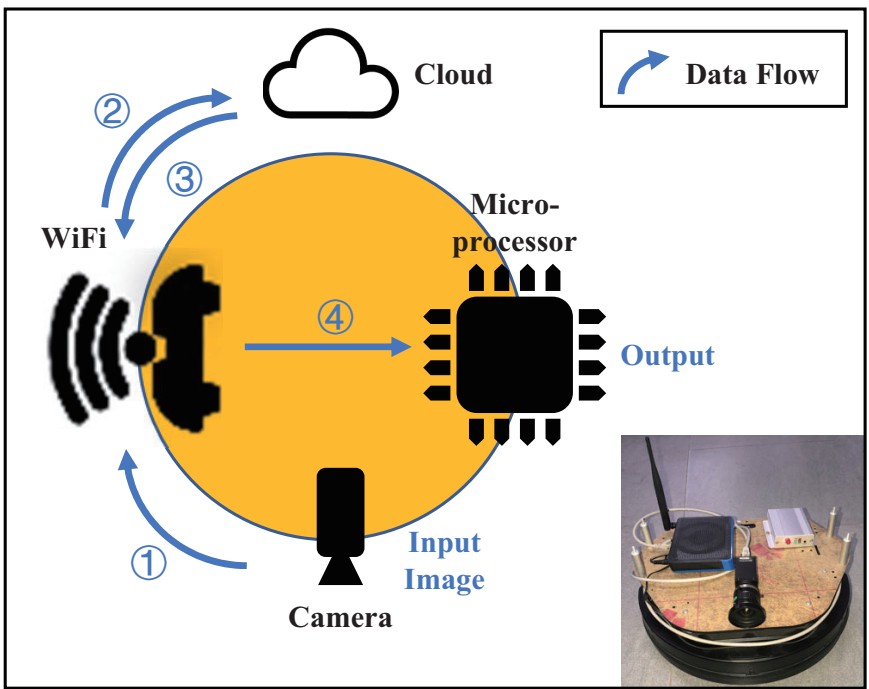

**Figure 8.** Turtlebot2 experimental platform.

**Table 3.** Experimental software environment.

| Software | Version |
| --- | --- |
| Operation System | Ubuntu 16.04 LTS , ROS Kinetic |
| Programming Language | Python 3.6.5 |
| Framework of Deep Learning | Tensorflow 1.13.0 |

*4.1. Garbage Target Detection Experiment*

We performed optimization work to improve the performance of the garbage detection model as the following:

(1) The garbage was subdivided into three types of labels, including paper, peel, and rubbish, to improve the accuracy of garbage identification.

(2) There was a severe imbalance among the data sample classes, which affected the learning of the SSD model. We used horizontal flip and vertical flip to double the sample size of the images of the peel tag category so that we could balance the ratio of the samples with the other two types.

(3) Data with a larger size can improve the neural network's performance effectively, and we used random channel offset and random noise addition techniques to augment the data.

We choose mean Average Precision (mAP) to measure garbage detection accuracy. The mAP is the average value of AP (Average Precision) for each category, as shown in Equation (4):

$$mAP = \frac{\sum\limits_{i=1}^{K} AP_i}{K},$$  (4)

where $K$ is the class of detected targets, and $AP$ denotes the average precision of each class of targets.

The classification of the experimental dataset and the performance of mAP on the test set are shown in Table 4, and the experimental parameters of each dataset are set as shown in Table 5.

In Table 4, the mAP value gradually increases on the three datasets, and the performance on each dataset during training is shown in Figure 9. The expansion and optimization of the dataset improves the balance between the samples, which is beneficial to fitting the detection model. The model appears to overfit after 20,000 steps when the dataset sample size is 310 and 484, and as the dataset sample increases to 1452, the model reaches its best-fit point after only 23,000 steps.

**Table 4.** Maximum accuracy performance on the test set for data sets of different sizes.

| Dataset | Dataset Size | mAP (%) |
|---|---|---|
| Original dataset | 310 | 73.45 |
| Sample-balanced dataset | 484 | 85.29 |
| Data augmented dataset | 1452 | 91.28 |

**Table 5.** Experimental parameters for each data set.

| Parameters | Value |
|---|---|
| Learn rate | 0.001 |
| Bitch size | 4 |
| Momentum | 0.9 |
| Ratio | 6:2:2 |

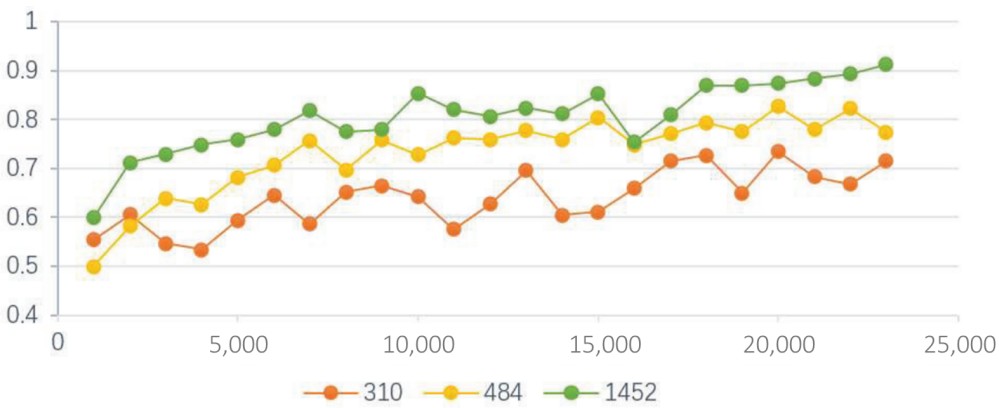

**Figure 9.** mAP of datasets with 310, 484, and 1452 samples under different iterations.

We selected the 1452 dataset and divided it into training, validation, and test sets in the ratio of 6:2:2. As a result, we can obtain the final ground trash detection model. After 23,000 steps, the Loss value stabilized, the model training was stopped, and the model reached the best fit and converged in Figure 10. The experimental result is shown in Figure 11. It can be seen that the garbage detection module has very high accuracy in detecting garbage such as paper, peel, and crumbs, and it is accurate in locating and identifying multiple targets in a single picture.

### 4.2. Indoor Depth Perception and Dynamic Local Path Planning Experiment

In order to validate the improved method of the self-encoder fully convolutional neural network, we collected indoor image and depth information and then we used the superpixel segmentation algorithm to discriminate the relative position and distance of obstacles based on the depth information. It is worth mentioning that we used the Microsoft Kinect 2.0 depth camera to collect the data. This camera is used to ensure the quality of the depth information as a way to verify the effectiveness of our improved algorithm. In the subsequent local dynamic path planning, the original camera Logitech C270c was used.

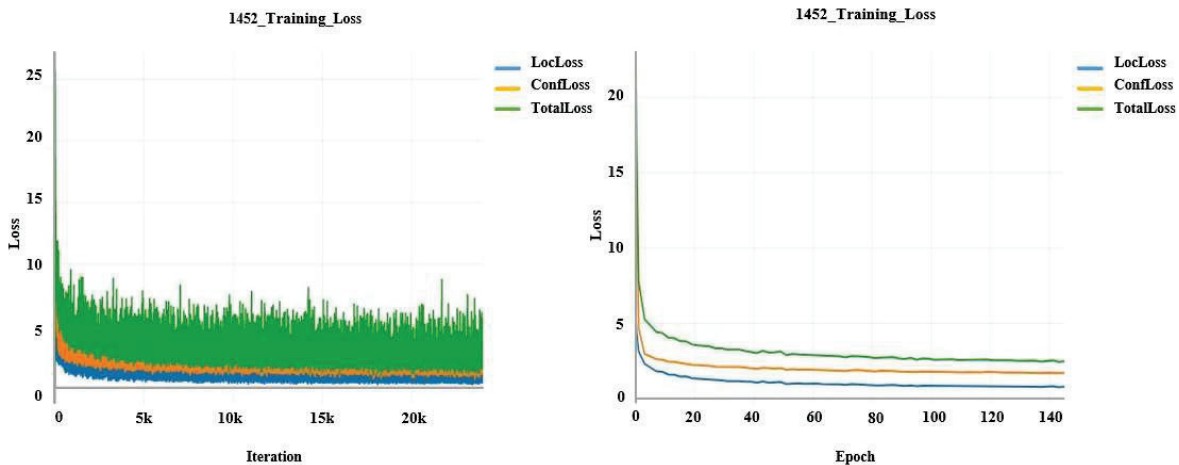

**Figure 10.** Variation in loss value for floor garbage detection.

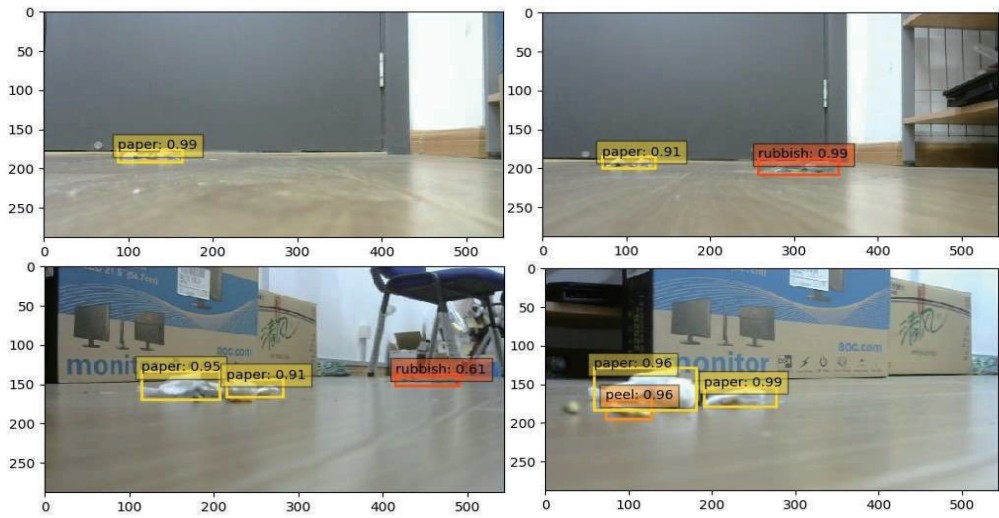

**Figure 11.** Indoor floor garbage detection experiment results.

### 4.2.1. Indoor Depth Perception

We acquired data using a Microsoft Kinect 2.0 depth camera, which contains a color camera and an infrared camera. The dataset has 1200 samples containing RGB images and depth images. Because of the installation position and field of view differences between the Kinect color camera and the infrared camera, the color images and the depth images need to be aligned. The camera calibration was performed using Zhang's calibration method [36]. The calibration platform used Matlab's Stereo Camera Calibration toolkit to calibrate the color and infrared images using a checkerboard grid image. Finally, it obtained the internal parameters and distortion coefficients and the rotation matrix and translation matrix of the two cameras. Finally, the depth map alignment results are shown in Figure 12.

In Figure 12, the depth map, shown in Figure 12b, has a large amount of noise and black hole areas, and there are still a small number of cases after alignment, Figure 12c. The anisotropic diffusion equation proposed by Vijayanager et al. [37] has a good repair effect of reducing the noise. The anisotropic diffusion equation restoration effect is shown in Figure 13.

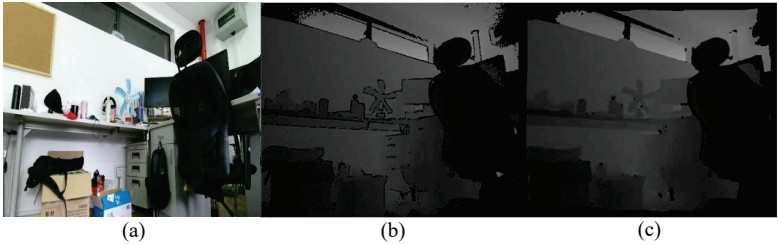

(a)                    (b)                    (c)

**Figure 12.** Kinect depth image alignment: (**a**) original scene image, (**b**) kinect acquisition depth map, (**c**) image alignment result.

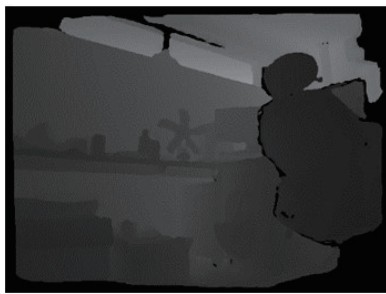

**Figure 13.** Anisotropic equation repair depth image.

Based on the improved method of the self-encoder fully convolutional neural network, the pre-training model of Laina et al. is experimentally selected to initialize the model parameters, which can accelerate the convergence of the model. After several experimental comparisons, the optimal parameters for this algorithm are as follows: the batch size is 4, the learning rate is initially 0.01, and the momentum is 0.9. After 70,000 steps, the model converged, and the algorithm model was obtained. The experimental results are shown in Figure 14, and the improved self-encoder convolutional neural network structure researched has a practical depth information estimation effect: (1) better handling of spatial dependencies and spatial distant and near connections of the scene; (2) the detection of contours between larger objects in space is evident; (3) the inference of the object depth is smoother and neater.

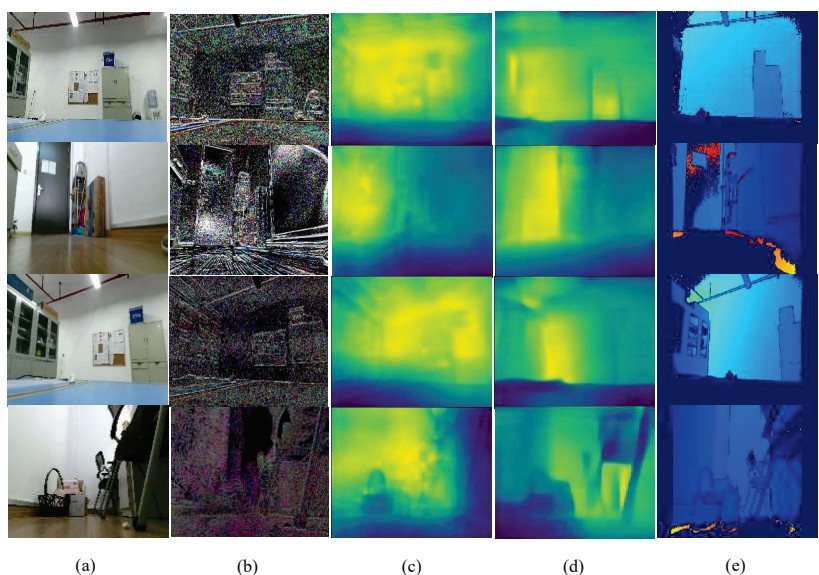

(a)            (b)            (c)            (d)            (e)

**Figure 14.** Experimental results of indoor environment depth information inference: (**a**) original scene picture, (**b**) parallax map, (**c**) Laina's inference result, and (**d**) this chapter's inference result (**e**) ground truth.

We used common metrics to assess experimental results:

Average relative error:

$$Rel = \frac{1}{T}\sum \frac{y_i^* - y_i}{y_i^*},$$ (5)

Root mean squared error:

$$RMS = \sqrt{\frac{1}{T}\sum (y_i^* - y)^2},$$ (6)

Average log10 error:

$$\log 10 = \frac{1}{T}\sum |\log y_i^* - \log y_i|,$$ (7)

Accuracy:

$$max(\frac{y_i^*}{y_i}, \frac{y_i}{y_i^*}) = \delta < threadhold.$$ (8)

The percentage of pixels to the total pixels needs to satisfy Equation (8), where $y_i^*$ and $y_i$ are the model's predicted and valid depth values of pixels, respectively, and $T$ is the sum of the number of test pixels.

Table 6 shows the result of Laina's method compared with the improved method based on the dataset containing 1200 images. Based on the self-encoding fully convolutional neural network proposed by Laina et al., we used inter-frame parallax maps to provide additional auxiliary information to the neural network. The accuracy improved by about 3.1% in the $\delta < 1.25$ accuracy metric, and the mean relative error Rel metric was reduced by about 10.4%.

**Table 6.** Comparison of experimental results on depth estimation.

| Method | Accuracy (Higher is Better) | | | Error (Lower is Better) | | |
|---|---|---|---|---|---|---|
| | $\delta < 1.25$ | $\delta < 1.25^2$ | $\delta < 1.25^3$ | RMS | Rel | log10 |
| Laina's [30] | 0.807 | 0.946 | 0.979 | 0.579 | 0.135 | 0.059 |
| Ours | 0.838 | 0.957 | 0.986 | 0.565 | 0.121 | 0.051 |

### 4.2.2. Dynamic Local Path Planning

Based on the depth information of the indoor scene, we used the superpixel segmentation algorithm to discriminate the relative position and distance of obstacles and determine the forward direction of the sweeping robot. For indoor scenes, human observation discriminated and marked the best forward direction, i.e., the non-obstacle area. The sample is shown in Figure 15.

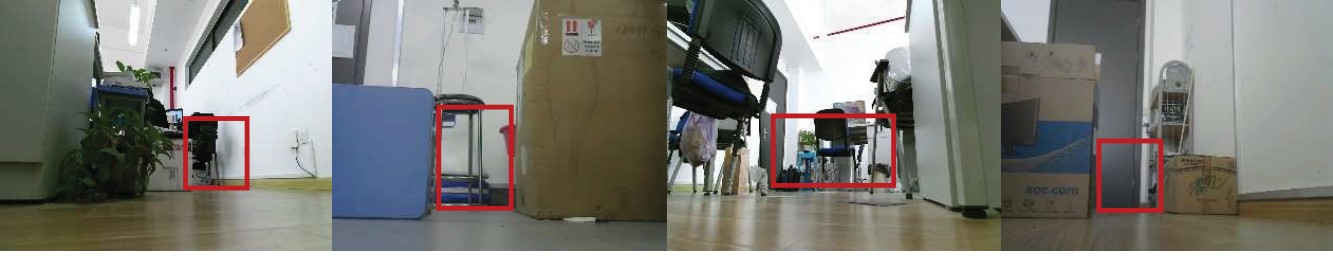

**Figure 15.** Indoor scene sample of path planning and forward direction calibration.

The experimental results are shown in Figure 16. The forward area is judged by the processing of the superpixel dynamic local path planning algorithm that matches the human-calibrated results. In the depth map obtained by SLIC, there is a clear difference

in the estimation results of the obstacle depth information at different distances, and the judgment of the non-obstacle area follows reality.

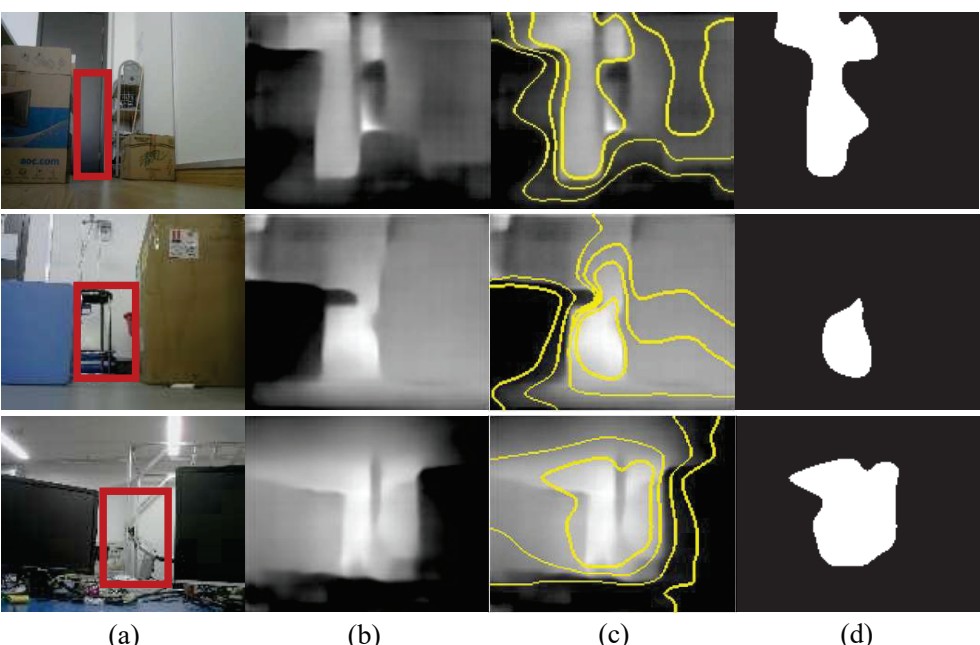

(a)        (b)        (c)        (d)

**Figure 16.** Superpixel dynamic path planning algorithm processing effect and sample comparison effect: (**a**) original scene map, (**b**) depth map, (**c**) superpixel segmentation result, and (**d**) average maximum superpixel block.

The superpixel dynamic local path planning algorithm focuses on dynamic processing, which requires a lot of the immediacy of path planning, and the processing speed must be much faster than the full coverage path planning. As shown in Figure 17, the average processing time is 12.554 ms, which can meet the requirement of immediacy well.

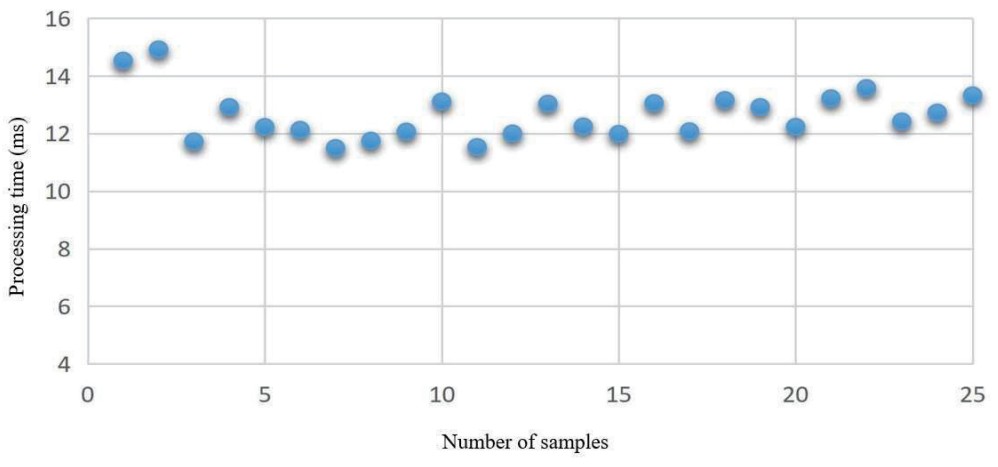

**Figure 17.** The processing time for 20 samples.

## 5. Conclusions and Outlook

To solve the problems of the high manufacturing cost, high hardware configuration requirements, and poor stability of existing sweeping robots, we proposed a technical framework of an autonomous visual navigation system based on a single camera and applied the framework to a sweeping robot, which eventually achieved the following expected results after experimental verification.

(1) The robot can realize the functions of garbage identification, environment depth information estimation, and superpixel dynamic local path planning.

(2) Based on the SSD algorithm, the robot can actively identify three types of garbage, such as paper, fruit peel, and rubbish. Its mAP accuracy reaches 91.28%, which meets the high-performance requirement. The active identification of garbage avoids the blind work of traditional visual navigation and improves the working intelligence of the robot.

(3) A modified method based on the self-encoder fully convolutional neural network is used to achieve monocular visual depth estimation. The depth information is used to solve the distance perception problem between the sweeping robot and the obstacles. The modified method improves the accuracy by about 3.1% when $\delta < 1.25$ and lowers the average relative error Rel by about 10.4%.

(4) We used the superpixel image segmentation algorithm to determine the obstacle area and the non-obstacle area. It takes only 12.554 ms to complete a single path planning, and the algorithm improves the efficiency of the navigation system and reduces the computational workload.

For the autonomous visual navigation system of the sweeping robot, its feasibility is verified through experiments. However, there are still many shortcomings and areas for improvement.

(1) The sweeping robot we researched only conducted experiments on light garbage, such as paper, peels, and rubbish. In the future, we will research the effects of more types of garbage on the sweeping robot to improve environmental adaptability.

(2) Our research avoided the drawback of over-reliance on environmental map modeling by traditional visual navigation techniques. However, there is still potential for improving the system stability, which can be optimized in the future to strengthen light anti-interference to make the path planning solution model more stable.

**Author Contributions:** Conceptualization, methodology, J.R. and H.B.; software, validation, formal analysis, H.B. and X.X.; investigation, writing—original draft preparation, H.B.; writing—review and editing, J.R. and J.C. All authors have read and agreed to the published version of the manuscript.

**Funding:** This research received no external funding.

**Institutional Review Board Statement:** Not applicable.

**Informed Consent Statement:** Not applicable.

**Data Availability Statement:** Not applicable.

**Conflicts of Interest:** The authors declare no conflict of interest.

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
