# Peer review of "Autonomous Visual Navigation System Based on a Single Camera for Floor-Sweeping Robot"

_applsci, doi:10.3390/app13031562_

Round 1

Reviewer 1 Report

This paper conceptualizes a new autonomous visual navigation system based on  single camera sensor for floor-sweeping robots. It investigates critical technologies such as floor  litter recognition, environmental perception, and dynamic local path planning based on depth maps. The system is applied to the TurtleBot robot for experiments, and the results show that the mAP  accuracy of the autonomous visual navigation system for fine trash recognition is 91.28%; it reduces  the average relative error of depth perception by 10.4% compared to conventional methods. Moreover, it has greatly improved the dynamics and immediacy of path planning.

The paper is well organized and the presentation of the work is good. 

However, I have the following concerns which are necessary to be considered in the revision to further improve the quality of the manuscript.

 Comments:

1.            Some more recent and relevant papers could be cited to support the literature review part of the paper like

 https://doi.org/10.3390/machines8030055

2.            Further clarification on the future work is needed.

3.            The quality of the Figs. 2 to 17 is not good. The Figures’ captions also need revision.

Reviewer 2 Report

The subject is interesting and is aligned with the readership and the themes of this journal. However, the paper does not include enough evidence to support the claim. The following bullet points include some suggestions to improve the manuscript to be publishable.

·       The paper must identify the gap that the research, which the authors conducted, fills.

·       Related work is too lengthy, but the more crucial issue is that the section does not have good organization. 

·       For readers to quickly catch your contribution, it would be better to highlight major difficulties and challenges, and your original achievements to overcome them, in a clearer way in abstract and introduction.

·       What are the other feasible alternatives? What are the advantages of adopting this technique over others in this case? How will this affect the results? More details should be furnished.

·       Some assumptions are stated in various sections. Justifications should be provided on these assumptions. Evaluation on how they will affect the results should be made.
 The discussion section in the present form is relatively weak and should be strengthened with more details and justifications.

·       There are some occasional grammatical problems within the text. It may need the attention of someone fluent in English language to enhance the readability.

·       I advise you just cite the papers in the literature review.

1.     Comparative Analysis of Low Discrepancy Sequence-Based Initialization Approaches Using Population-Based Algorithms for Solving the Global Optimization Problems

2.     A modified bat algorithm with torus walk for solving global optimization problems

·        Firstly, for section 1, authors should provide more specific comments of the cited papers after introducing each relevant work. What readers require is, by convinced literature review, to understand the clear thinking/consideration why the proposed approach can reach more convinced results. This is the very contribution from authors. In addition, authors also should provide more sufficient critical literature review to indicate the drawbacks of existed approaches, then, well define the main stream of research direction, how did those previous studies perform? Employ which methodologies? Which problem still requires to be solved? Why is the proposed approach suitable to be used to solve the critical problem? We need more convinced literature reviews to indicate clearly the state-of-the-art development.

Round 2

Reviewer 1 Report

Well Done!